# Mechanical and Corrosion Behaviour in Simulated Body Fluid of As-Fabricated 3D Porous L-PBF 316L Stainless Steel Structures for Biomedical Implants

**DOI:** 10.3390/jfb15100313

**Published:** 2024-10-21

**Authors:** Pedro Nogueira, João Magrinho, Luis Reis, Augusto Moita de Deus, Maria Beatriz Silva, Pedro Lopes, Luís Oliveira, António Castela, Ricardo Cláudio, Jorge L. Alves, Maria Fátima Vaz, Maria Carmezim, Catarina Santos

**Affiliations:** 1IDMEC—Institute of Mechanical Engineering, Instituto Superior Técnico, Universidade de Lisboa, 1049-001 Lisboa, Portugal; pedroafonsonogueira2@hotmail.com (P.N.); joao.magrinho@tecnico.ulisboa.pt (J.M.); luis.g.reis@tecnico.ulisboa.pt (L.R.); beatriz.silva@tecnico.ulisboa.pt (M.B.S.); 2CeFEMA—Center of Physics and Engineering of Advanced Materials, Instituto Superior Técnico, Universidade de Lisboa, 1049-001 Lisboa, Portugal; amd@tecnico.ulisboa.pt; 3INEGI, Faculty of Engineering, University of Porto, 4200-465 Porto, Portugal; pmlopes@inegi.up.pt (P.L.); loliveira@inegi.up.pt (L.O.); falves@fe.up.pt (J.L.A.); 4Instituto Politécnico de Setúbal, EST Setúbal, Campus IPS, 2910-761 Setúbal, Portugal; antonio.castela@estsetubal.ips.pt (A.C.); ricardo.claudio@estsetubal.ips.pt (R.C.); 5CQE, IMS, Instituto Superior Técnico, Universidade de Lisboa, Av. Rovisco Pais, 1049-001 Lisboa, Portugal

**Keywords:** laser powder bed fusion, 316L stainless steel, micro-CT porosity, corrosion resistance, SBF, mechanical properties, biomedical implants

## Abstract

Laser powder bed fusion (L-PBF) is one of the most promising additive manufacturing technologies for creating customised 316L Stainless Steel (SS) implants with biomimetic characteristics, controlled porosity, and optimal structural and functional properties. However, the behaviour of as-fabricated 3D 316L SS structures without any surface finishing in environments that simulate body fluids remains largely unknown. To address this knowledge gap, the present study investigates the surface characteristics, the internal porosity, the corrosion in simulated body fluid (SBF), and the mechanical properties of as-fabricated 316L SS structures manufactured by L-PBF with rhombitruncated cuboctahedron (RTCO) unit cells with two distinct relative densities (10 and 35%). The microstructural analysis confirmed that the RTCO structure has a pure austenitic phase with a roughness of ~20 µm and a fine cellular morphology. The micro-CT revealed the presence of keyholes and a lack of fusion pores in both RTCO structures. Despite the difference in the internal porosity, the mechanical properties of both structures remain within the range of bone tissue and in line with the Gibson and Ashby model. Additionally, the as-fabricated RTCO structures demonstrated passive corrosion behaviour in the SBF solution. Thus, as-fabricated porous structures are promising biomaterials for implants due to their suitable surface roughness, mechanical properties, and corrosion resistance, facilitating bone tissue growth.

## 1. Introduction

Stainless Steels (SS) can be used in biomedical implants, especially in screws and fixations used in bone healing, due to their excellent malleability, robust mechanical properties, good corrosion resistance, biocompatibility, and cost-effectiveness [1,2].

There is a need for biomedical implants to be patient-specific and customised in design. Traditional manufacturing methods fall short of providing the capacity required to produce complex designs and personalised features essential for biomedical implants. Additive manufacturing (AM) is an important and emergent set of technologies capable of producing customised bone healing implants, like bone scaffolds, intramedullary rods, and osteosynthesis plates, according to the patient’s requirements, with a graded porous complex design that aims to mimic trabecular and cortical natural bone tissue [3,4]. The natural porosity of trabecular bone leads to significantly lower values of mechanical properties, namely the elastic modulus, which presents values in the range of a few tenths of GPa [5]. In addition to design freedom, AM allows the production of highly porous metallic structures with low elastic modulus, capable of overcoming the detrimental stress shielding effect that currently occurs with permanent bulk metallic implants [6]. Also, AM 316L stainless steel highly porous scaffolds are possible candidates for repairing trabecular bone defects. These porous scaffolds exhibit mechanical properties (elastic modulus and yield strength) close to those of the trabecular bone and cytocompatibility (after one day) that is close to wrought 316L stainless steel [7]. In addition to a low elastic modulus and an adequate biological response, other determining factors ensure the high performance of AM 316L SS structures, such as corrosion resistance [8].

The AM 316L SS microstructure, mechanical and corrosion behaviour, are affected by additive manufacturing processing parameters, like laser beam power and scanning speed, as well as cooling rate and directional solidification [9].

The distinctive microstructure and superior corrosion behaviour of 316L stainless steel fabricated by laser powder bed fusion (L-PBF) have been widely reported and become consensual. In comparison with other additive manufacturing techniques, laser powder bed fusion (L-PBF) 316L SS might surpass the conventional wrought 316L SS in terms of corrosion resistance and biological response [10].

Moreover, laser powder bed fusion L-PBF 316L SS pitting resistance behaviour was attributed to the absence of large microstructural MnS inclusions and passive oxide film semiconducting properties, with a lower concentration of oxygen vacancies. Also, the L-PBF-316L microstructure included fine sub-grains inside the grain structures [11].

L-PBF 316L SS parts manufactured under high laser power (220 W) showed improved biocompatibility, higher pitting potential, and lower corrosion rate in the simulated fluid body (SBF) compared with selective laser melting (SLM) 316L formed at low laser power. Concomitantly, the passive film formed on L-PBF 316L presents a higher thickness but less Mo content than on similar quenched 316L [12].

In all the works reported above, a post-process surface modification was carried out. It is well known that these post-processes increase the price and production time, as they require additional equipment and an extra step. In addition, they can introduce new defects into the structures, as well as not allow a uniform surface due to the complexity of the geometry. No less important is the contribution to environmental impact, which can be negative because chemical reagents or other materials that have safety concerns are commonly used [13]. Having this in mind, this study investigated the mechanical and corrosion behaviour of 316L SS porous 3D structures as-fabricated (without surface finishing) composed of RTCO unit cells and two relative densities, produced by laser powder bed fusion. With a biomedical implant application in view, the two selected structures’ relative densities (10% and 35%) attempt to mimic cortical and trabecular bone densities, respectively, and are seen to achieve mechanical behaviour similar to the one found in natural bone. The aim is also to ensure that the fabricated 3D structures maintain the characteristics of biocompatibility and promote the regeneration of bone tissue.

## 2. Materials and Methods

### 2.1. RTCO Structures Design

Lattice structures composed of Rhombitruncated Cuboctahedron (RTCO) unit cells were selected to be designed and produced by laser powder bed fusion (L-PBF) from a 316L SS to evaluate their microstructure, corrosion behaviour, and compressive properties.

The RTCO lattices were designed in Solid Edge 2020 (SIEMENS, Munich, Germany). The specimens used for the compression tests were cylindrical in shape, with a diameter of 35.0 mm, a height of 35.0 mm, and at least 10 cells in diameter, in accordance with ISO 13314 standards [14]. The 3D structures used to evaluate microstructure and biodegradation behaviour were fabricated with a diameter of 17.5 mm, utilising the same RTCO lattice design. The relative density of the fabricated structures, which is evaluated by the volume occupied by the 3D structure divided by the total volume, was chosen to be 10% and 35%. The 3D structures with a 10% density will be referred to as RTCO10, while those with a 35% density will be called RTCO35. Figure 1 shows a schematic representation of the 316L SS structures with RTCO unit cells with two different relative densities, manufactured by L-PBF.

### 2.2. Production of 316L SS RTCO Structures by Laser Powder Bed Fusion

The 3D RTCO structures were manufactured using 316L stainless steel (316L SS) powder, sourced from GE Additive, New York, US. In this experimental setup, several 3D structures with 10% and 35% density and bulk with the same dimensions were manufactured on a Concept Laser M2 Series 5 3D Printer (GE Additive, New York, NY, USA), using 316L SS powder from the GE supplier [15].

The process parameters were meticulously set to ensure optimal print quality: laser power was calibrated to 300 W, scanning speed adjusted to 700 mm/s, with a layer thickness of 50 µm, a spot size (hatch distancing) of 130 µm, and an energy density (*Ev*) of 66 J/mm^3^, determined according to the equation.
(1)$$Ev=\frac{p}{v\cdot h\cdot t}$$
where p is the laser power (W), v is the scanning velocity (mm/s), h is the hatch distancing (mm), and *t* is the layer thickness (mm) [16].

To mitigate potential issues with the removal of the job from the built plate and enhance structural support, each 3D RTCO structure was produced over supports with a 5 mm height. Additionally, to optimise the printing process and reduce the likelihood of overheating and bending, a 5-degree orientation angle on the x-axis along with block-type support structures was systematically incorporated into each structure’s design. This approach underscored the rigorous scientific methodology applied to enhance the fidelity and structural integrity of the printed objects.

After printing, the 3D RTCO structures were carefully removed from the build plate with the help of a metal nipper. The quality of each structure was visually analysed, and global dimensional accuracy was assured using a digital calliper. Furthermore, the platforms printed under the 3D RTCO structures were removed by electrical discharge machining (EDM) on a CHARMILLES Robofil 190 machine.

### 2.3. Physico-Chemical Characterisation of 316L SS 3D RTCO Structures

The morphological characterisation of the 3D RTCO structures was conducted on a benchtop scanning electron microscopy (SEM), ThermoScientific model Phenom ProX G6, equipped with a CsB6 filament, with the elemental chemical composition determined by the respective X-ray energy dispersive spectrometer (EDS). The 3D micro-architectural morphology and porosity of each 3D RTCO structure were characterised by X-ray microtomography (micro-CT), using a Phoenix V|TOME|X, GE. μCT. For this purpose, 3D RTCO structures were scanned at a voltage of 80 kV with a current intensity of 80 mA. The scanning angular increment was 0.15°, and the spatial resolution was 6.7 μm. Acquired image data were qualitatively and quantitatively interpreted using 3D tomographic reconstruction and analysis software (Volume Graphics 3.04 software, Volume Graphics). The sphericity (S) of the pores measured by micro-CT was defined by the ratio of the surface area of a sphere (with the same volume as the defect, (*V*) to the actual surface area of the defect (*A*) [17]. The sphericity can be expressed as follows:(2)$$S=\frac{{\left(6V\right)}^{2/3}\cdot \pi^{1/3}\;}{A}$$

The crystalline structure of 316L SS powder and 3D RTCO structure were identified by X-ray diffraction (XRD) using a D8 Advance Bruker AXS θ–2θ diffractometer with a copper radiation source (Cu Kα, λ = 1.5406 Å) and a secondary monochromator operated at 40 kV and 40 mA over a 2θ range of 10° to 90°, with a step size of 0.02° and a step time of 0.6 s per step.

Due to the complexity of the geometry, it is difficult to manufacture every strut to its exact dimensions in the lattice structure. This difficulty results in additional material irregularly distributed throughout the various struts. Also, during the manufacturing process, some metal powder is trapped inside the lattice. To remove the excess powder, the 3D RTCO structures were vibrated using ultrasound for 20 min in a 97% ethanol solution, ensuring that most of the trapped 316L SS unsintered powder was removed during the cleaning procedure. The relative density of the 3D RTCO structures was experimentally determined after cleaning by using the weight method [18]:(3)$$\bar{\rho }=\frac{\rho_{c}}{\rho_{m}}=\frac{W_{specimen}}{V_{bulk}.\rho_{bulk}},$$
where $W_{specimen}$ is the mass of the structure, $V_{bulk}$ is the volume of the cylinder with the outer dimensions of the structure, and $\rho_{bulk}$ is the density of the 316L SS. The relative density, measured by the weight method for the three 3D RTCO structures, with the two theoretical relative densities (10% and 35%) are shown in Table 1.

### 2.4. In Vitro Biodegradation of 3D RTCO Structures Under Simulated Body Fluid Conditions

Electrochemical characterisation, including open circuit potential (OCP) measurements and potentiodynamic polarisation tests, was conducted. OCP measurements were taken for all structures in simulated body fluid (SBF) at pH 7.4 and 37 °C prior to performing the potentiodynamic polarisation tests. The active/passive behaviour was determined by potentiodynamic polarisation using both porous structures (RTCO10 and RTCO35) and bulk L-PBF immersed in simulated body fluid (SBF) at pH 7.4 and 37 °C [19]. With this aim, potentiodynamic curves were obtained by polarisation in the anodic direction, with a scan rate of 1 mV/s. Measurements were performed using a three-electrode electrochemical cell with a saturated calomel reference electrode (SCE), a platinum counter electrode, and a potentiostat Gamry Instruments Reference 600+. The bulk was metallographically prepared using silicon carbide paper by surface grinding with water refrigeration up to a 2500-grit surface finish. All RTCO structures and bulk were surface cleaned with acetone and air flux dried.

### 2.5. Experimental Compression Tests

The compression tests featured three cylindrical bulk L-PBF and six 3D RTCO structures, three of each relative density, 10% and 35%. All the compression tests were carried out in an Instron SATEC 1200 with a load cell of 1200 kN at a constant speed of 2.5 mm/min, according to the standard ISO 13314 [14]. The top and bottom of the 3D RTCO structures were covered with a thin layer of disulphide molybdenum (MoS_2_) grease to ensure a low friction coefficient between the specimens and the compression plates. The quasi-elastic slope (Young’s modulus) was calculated from the results of the compression tests.

### 2.6. Finite Element Simulations of Compression

A finite element model was created to replicate the laboratory conditions. In the finite element analysis, the 3D RTCO structures were compressed between two rigid surfaces in a dynamic analysis (ABAQUS/Standard) at the same speed of 2.5 mm/min. The interaction between the 3D RTCO structures and the testing plates was modelled with a friction coefficient of 0.05 [20]. A mesh convergence analysis was performed, leading to an average mesh size of 0.25 mm for both RTCO geometries. In the finite element analysis, only one-quarter of the RTCO geometry was used, and symmetry boundary conditions were applied to save computational time.

The 3D RTCO structure mechanical properties considered in the simulations were retrieved from a previous study by the authors where bulk compression tests were done on structures produced in the same machine and the same conditions [21]. The Young’s modulus and yield stress considered for the bulk structure were 195 GPa and 449 MPa, respectively [21].

## 3. Results and Discussion

### 3.1. Physicochemical Characteristics of 3D RTCO Structures

The fresh 316L stainless steel powder predominantly exhibited a spherical shape with small satellites attached to some larger 316L stainless steel powder, as represented in the inset in Figure 2A, with an average size of 21.2 ± 6.0 μm, distributed in a Gaussian fashion ranging from 5 to 50 μm (Figure 2B). The elemental composition of the 316L SS powder, determined through EDS analyses, is presented in the inset of Figure 2B. The obtained composition of fresh 316L SS was similar to that provided by the manufacturer [15].

The X-ray diffraction (XRD) spectrum of the 316L SS precursor powder depicted in Figure 3 confirmed the presence of a pure austenitic phase. This was evidenced by distinct crystallographic peaks corresponding to γ(111), γ(200), γ(220) and γ(311) orientations at (2θ) angles of 44.08°, 51.24°, 75.08°, and 90.94°, respectively. Upon fabrication of the 3D RTCO10 structures via the L-PBF process, similar crystallographic peaks were observed, confirming the dominance of a pure face-centred cubic (fcc) gamma (γ) austenite phase with no discernible secondary ferrite peaks (Figure 3). According to the literature, secondary delta (δ) ferrite peaks may arise due to the spattering effect during the L-PBF process, particularly when using recycled powder [22]. The exclusive use of fresh 316L SS powder in the fabrication of the 3D RTCO structures ensured the absence of any ferrite phase. Moreover, it has been documented that the laser beam power significantly affects the formation of the ferrite phase during the L-PBF process. Notably, the ferrite phase may emerge under considerably higher laser beam powers, such as 500 W [12,23]. Considering that in the L-PBF additive manufacturing process, 300 W laser power was used, it was anticipated that the formation of the ferrite phase would be prevented. Additionally, it was noted that in fresh 316L SS powder and the 3D RTCO10 structures, the most prominent XRD peak corresponds to the (111) plane. Previous studies have highlighted that the scanning strategy employed during the fabrication and build direction dictated this crystallographic orientation [24,25]. For instance, 3D structures constructed with a 45° build direction typically exhibit a preference for the (111) crystallographic orientation, while those with a 90° build direction (vertical build direction) show a preference for the (220) plane [25]. The XRD pattern of the 3D RTCO10 structures obtained is similar to the results reported in the literature [25]. Additionally, it can also be noted that in comparison to 316L SS powder, a certain broadening of the peaks in the 3D RTCO10 structures was observed. This broadening can be attributed to residual stresses and lattice distortion induced by the L-PBF process [25,26].

It is well known that scan strategy and process parameters could significantly impact the dimensional and geometrical accuracy, as well as the surface roughness of produced 3D RTCO structures [27]. Having this in mind, scanning electron microscopic analyses were conducted on RTCO10 and RTCO35 to evaluate their dimensional and geometrical accuracy and surface roughness, along with their top surface morphology. Figure 4 shows the top and front view of optical photographs as well as the secondary electrons (Figure 4a,b,d; Figure 4g,h,j) and backscattered images (Figure 4c,i) of the 3D RTCO10 and RTCO35 manufactured structures. SEM images show that no significant differences were observed in the two RTCO structures. However, it is possible to observe in Figure 4j,i of the RTCO35 the morphology of the melting pool. The width of the melt pool was measured at approximately 180 µm. Moreover, it can be seen in Figure 4d,j that the layer thickness of the structures produced by L-PBF is 300 μm and 600 μm for RTCO10 and RTCO35, respectively. These measured layer thicknesses are greater than those designed in CAD, which were 230 μm for RTCO10 and 490 μm for RTCO35. This geometric difference between the theoretical CAD design and the actual printed part was expected [28,29]. This discrepancy is attributed to the layer-by-layer build-up during additive manufacturing, known as the stair-step effect, which is more pronounced on inclined surfaces [30]. To reduce this effect, it is proposed to use starting powders with a smaller diameter along with a thinner layer thickness [28,29,30]. Moreover, for optimal layer quality, the nominal layer thickness of two to three times the maximum powder particle diameter is recommended [31].

As shown in Figure 4e,f,k,l, the measured top surface roughness (Ra and Rz) of the RTCO10 and RTCO35 structures, determined using 3D Roughness Reconstruction software, was approximately 20 μm for Ra and 8 μm for Rz for both RTCO structures. This indicates that an increase in RTCO density does not significantly influence the surface roughness of the manufactured structures. Additionally, it can be observed that the surface roughness (Ra) is comparable to the size of the fresh 316L powder. Surface roughness (Ra) values of 316L structures manufactured by L-PBF have been reported to range from 10 to 56 μm, depending on the process parameters [32,33]. For instance, Chen et al. [34] found that higher laser power, increased traverse speed, and lower powder feed rate resulted in a lower surface roughness (Ra). The finer roughness observed was attributed to the presence of partially melted embedded 316L particles, as shown in Figure 4. Although the presence of these partially melted embedded 316L particles has been reported to have a negative impact on the corrosion performance and mechanical properties of the L-PBF structures [9,35], these surface characteristics, particularly roughness, are crucial in bone healing applications. Rough surfaces can influence microbial adhesion and subsequent biofilm development [36]. Additionally, increased surface roughness and topography can promote cell adhesion and growth due to increased surface area [37].

It is well documented that 3D structures produced by AM are prone to being laden with volumetric defects [31,38]. These defects, commonly identified as lack of fusions (LoFs), gas-entrapped pores (GEPs), and keyholes (KHs), act as stress enhancers, which can significantly compromise the mechanical properties of the produced 3D structures [31,38,39]. To evaluate the volumetric defects of both 3D RTCO structures, µCT analyses were performed. Figure 5 and Figure 6 show 3D reconstructions of µCT images of the 3D RTCO structures manufactured with 10% and 35% relative densities, respectively. In Figure 5a,b, it can be observed the 3D morphology of the 3D RTCO10 structure with 10% relative density as well as the distribution of the porosity throughout the 3D structure, scanned by µCT. The blue and green points observed in the bottom view of Figure 5b and magnified in Figure 5f,g are pores detected inside the RTCO10 structure. The volume of the blue pores ranges from 0.0001 to 0.0002 mm^3^, while the green pores range from 0.0004 to 0.0008 mm^3^. Nevertheless, these larger pores are fewer in number and are uniformly distributed throughout the 3D structure. However, in the top view (Figure 5), large pores (volume ~0.0010 mm^3^) are also visible. M. Mahmood et al. produced 3D 316L SS structures with a hatch distance of 120 µm and pores volume of 0.0005 mm^3^. Given the hatch distance used in this work (130 μm), it is likely that the pores’ volumes are of a similar magnitude [40].

In relation to the RTCO35 structure (Figure 6), mCT analyses indicate a lower number of pores (identified in blue in Figure 6c,f,g) compared to the RTCO10 structure (Figure 5).

The shape and orientation of these pores (Figure 5 and Figure 6), characterised by their sphericity, are displayed in Figure 7. For the RTCO10, most pores exhibit a mean aspect ratio ranging between 0.6 and 0.7, which is consistent with ellipsoidal pores [41]. Nonetheless, a few pores with flat shapes (sphericity ~0.45) and others with more spherical shapes (sphericity ~0.75) are also observed. The presence of this diversity in sphericity is attributed to the experimental parameters employed in the L-BPF process. Therefore, it appears that with an increase in the relative density (RTCO 35%), the shape of the average pores remains relatively unchanged, as observed in Figure 7.

According to the literature [42], most L-PBF processing occurs in the keyhole regime, but before the onset of keyhole instabilities that lead to pore formation. Overall, the median pore radius remains below 0.04 mm, as depicted in Figure 7.

Based on the CT scans, further information about the size and shape of the pore defects of both RTCO structures can be obtained, as shown in Figure 7. In Figure 7a, the pore diameter is plotted against the sphericity of the pore’s defects. As illustrated, there is a difference in the diameter and sphericity of the pores between the RTCO10 and RTCO35 structures. Although the porosity of RTCO10 is higher, both 3D structures lack large and irregular defects. Additionally, it can be observed that pores with a larger diameter tend to be more aspherical in both structures. The pores in RTCO35 tend to be smaller (Figure 7c) than those in RTCO10 (Figure 7b) and are more spherical, which supports the assumption that they might be keyhole pores. The presence of entrapped gas pores cannot be determined since the resolution of the CT scans is not sufficient to detect these pores’ defects.

To validate the results from the µCT analyses, SEM analyses were performed. The images are presented in two orientations, namely normal and parallel to the building direction. As depicted in the low magnification SEM image (Figure 8a), areas with lack of fusion are evident at the top of the rhombitruncated cuboctahedron, with a size comparable to the average size of the 316L powder used in the manufacturing process. Conversely, in the bottom view, apart from the lack of fusion, voids with spherical morphology (aspect ratio close to 1) and size smaller than 10 μm are detected. The presence of these voids, known as spherical-shaped gas pores, indicates insufficient time for gaseous bubbles to rise to the top of the melt pool and be released before solidification during the L-PBF process [43]. The occurrence of such defects is attributed to the fluid dynamics within the molten pool, influenced by buoyancy force, Lorentz force, surface tension gradient, temperature gradient, recoil pressure, and/or laser-induced shear stress [41]. The presence of these defects, including pores and voids, significantly compromises the quality of the 3D RTCO structures and can notably diminish mechanical properties like yield strength.

Examining in detail the interior of one lack of fusion pore (Figure 8d), a fine cellular morphology with an average cell size of approximately 0.76 ± 0.30 μm was observed. Similar average cell sizes have been reported by other authors [44,45]. It is known that cell size is influenced by solidification conditions such as thermal gradient, cooling rate, and solidification front velocity, which are determined by the process parameters and component geometry. Faster solidification leads to a finer cellular or dendritic structure [44]. However, there are different theories regarding the formation sequence of these cellular structures, and there are currently gaps in knowledge about how the cellular structure is formed.

Although 316L SS parts produced by additive manufacturing techniques have been studied, there is a lack of deeper understanding regarding their corrosion-related properties in simulated body fluid solutions for biomedical implants. Additionally, the literature shows contradictory results, namely when comparing such to conventionally produced counterparts, indicating both increased and reduced pitting resistance [46,47,48]. Furthermore, it is recognised that surfaces with reduced roughness generally exhibit more noble pitting potentials [49,50]. Another aspect known to influence corrosion resistance is the presence of porosity related to a Lack of Fusion (LOF), which typically decreases pitting corrosion resistance [51], although in general, the pitting potential did not vary significantly with other typical porosities [50]. However, this porosity was not detected on the surface of the RTCO structures, so it is believed that it will not impact their corrosion resistance. Given that the RTCO structures have similar roughness but exhibit different densities and internal non-uniform LOF porosity, potentiodynamic polarisation measurements were performed to evaluate their pitting corrosion resistance.

The potentiodynamic polarisation (CP) curves of Figure 9 show the comparison between the corrosion behaviour of polished bulk L-PBF and the two non-surface finishing RTCO structures (10% and 35%) in SBF solution, containing Cl^−^ ions. The Bulk CP curve presents a Tafel-like region before the passivation plateau, followed by an abrupt current enhancement associated with a local breakdown of passive films. This behaviour is similar to the ones found in literature [52]. The L-PBF bulk structure presents a passive plateau of ~500 mV, a pitting potential of about +676 mV (vs. SCE), and a passive current density of 612 mA/cm^2^. However, the passive plateau is shorter for bulk L-PBF, revealing that the surface passive films are still protective but for lower potentials. The RTCO10 structure presents a pitting potential of about Epit +342 mV (vs. SCE) and RTCO35 a pitting potential of Epit +378 mV (vs. SCE), being the passive current density about 612 mA/cm^2^ and 610 mA/cm^2^, respectively. The bulk repassivation potential corresponding to pit repair by a passive layer is about +100 mV and more noble than the repassivation potential of RTCO structures. Repassivation behaviour is also similar for both RTCO (10% and 35%) structures, with a repassivation potential of Erep +61.2 mV (vs. SCE) and Erep +29.1 mV (vs. SCE), respectively. Moreover, for 3D structures after passive film breakdown, the current enhancement is not as abrupt as usual, but there is a significant current density increase with potential suggesting the effect of the porous lattice. The differences between RTCO10 and RTCO35 lattice structures observed in Figure 9 do not seem to induce major changes.

### 3.2. Compression Tests

The experimental stress-strain curves of the RTCO specimens are presented in Figure 10. All specimens exhibited a stress-strain curve with a linear region that smoothly transitions to the plastic regime, reaching a plateau in the case of specimens with a 10% relative density and a steadily increasing region in those with a 35% relative density. At the end of the curves, all specimens exhibited a densification region, where the struts of the lattice have come in contact, greatly increasing the load that must be applied to continue deformation. These three stages of deformation have been reported by several authors and are typical of bending-dominated lattices, like those made of RTCO unit cells [53,54,55,56].

For both geometries, the results of the three specimens are similar, except for the RTCO10 S3. This specimen initiated its plastic deformation earlier, likely due to small defects in the geometry.

In Figure 10c,d, both the experimental and the computational (finite element) stress-strain curves are exhibited. For both geometries (RTCO10 and RTCO35), the overall shape of the computational stress-strain curve (Figure 10c,d) is similar to the experimental ones (Figure 10a,b).

Young’s modulus is very similar between experimental and computational data in the geometry with a relative density of 35%, as is reported in Table 2. However, the difference is notorious in the geometry with a relative density of 10%, with Young’s modulus of the experimental results being, on average, 36.9% higher than the computational results.

The experimental curves (Figure 10a,b) display a higher yield stress and overall higher stresses during the plastic regime when compared with the numerical stress-strain curves presented in Figure 10c,d. This is because the numerical modelling considered the theoretical relative density value instead of the real value. The experimental RTCO structures have a higher relative density than designed (see Table 1). This extra material, due to the differences in strut dimensions described previously and the presence of embedded partially melted 316L particles, plays a key role in deformation after yielding, with its role being diminished in the elastic regime. The extra material can be present in two different regions: randomly distributed in the struts, or the joints/corners of the geometry, as represented in SEM images (Figure 4) and microCT (Figure 5 and Figure 6). The randomly distributed partially melted embedded 316L particles of the struts are not load-bearing until densification, unless it is in large quantities, making its distribution very uniform in most struts, acting like an increase in thickness, as observed in Figure 4. The extra particles on the joints/corners are load-bearing and influence the mechanical response, in particular after yielding. Once the plastic regime is approached, the struts that are supported in the corners begin to bend, and the extra partially melted 316L particles on the corners provide additional strength that is not contemplated in the finite element model. This effect, coupled with the difference in strut dimensions, results in a higher-than-expected yield stress.

To illustrate this, Figure 10d highlights how RTCO35_S6, the 3D structure with a slightly higher relative density, exhibits higher stresses than the others and how this difference is accentuated as deformation increases. Despite this, Young’s modulus of this RTCO35_S6 structure (17498 MPa) is not much higher than the ones of the other two geometries (RTCO35_S4-16229 MPa and RTCO35_S5-15208 MPa); it is approximately as much above average as RTCO10_S2 is below average. The geometry with a relative density of 10% was produced with approximately 18%, which corresponds to an increase of 8%. When this gain in density occurs, some of the randomly distributed particles increase the thickness, as observed in Figure 4d, thus justifying not only the difference in the plastic regime but also the difference in Young’s modulus, as claimed by other authors [18], who reported a 6.8% increase in density for lattices designed with 20% relative density [18]. Despite being a very important factor, the effect of these differences in relative density is not the only factor that influences the mechanical response. The defects and imperfections inherent to the LPBF process also play an important role that has been highlighted in the literature [57,58].

The Gibson and Ashby model of an open cell structure may be applied and compared with the results obtained. The two equations that relate the relative Young’s modulus and the relative yield strength with the relative density $\frac{\rho_{c}}{\rho_{m}}$ are as follows:(4)$$\frac{E_{c}}{E_{m}}=C_{1}{\left(\frac{\rho_{c}}{\rho_{m}}\right)}^{n_{1}}$$
(5)$$\frac{\sigma_{c}}{\sigma_{m}}=C_{2}{\left(\frac{\rho_{c}}{\rho_{m}}\right)}^{n_{2}}$$
where $E_{c}$ and $\sigma_{c}$ are Young’s modulus and yield stress of the lattice structure, $E_{m}$ and $\sigma_{m}$ are Young’s modulus and yield stress of the base material and $\;\rho_{c}$ and $\rho_{m}$ are the density of the lattice structure and the density of the base material, respectively. $C_{1}$, $C_{2}$, $n_{1}$, and $n_{2}$ are just fitting constants [56].

The results of the relative density vs. the relative Young’s modulus or the relative density vs. the relative yield strength were plotted in Figure 11a,b, respectively. In Figure 11a, two straight lines were drawn, corresponding to $n_{1}=2$, $C_{1}=0.25$ and $n_{1}=2$, $C_{1}=1$, and in Figure 11b the straight lines correspond to $n_{2}=1.5$, $C_{2}=0.25$ and $n_{2}=1.5$, $C_{2}=1$. These lines can be regarded as interpolations of the data. The grey dots indicate the simulation results obtained with the designed relative density, and the orange dots indicate the experimental results with the manufactured relative density. The results suggest that the RTCO cell type is bending-dominated, as the slopes of the blue straight lines follow what is reported by Ashby’s model: bending-dominated structures have coefficients with values close to $n_{1}=2$ and $n_{2}=1.5$ [56].

## 4. Conclusions

This study was designed and carried out to evaluate the impact of as-fabricated 3D 316L SS structures, produced using L-PBF technology without surface finishing at two different densities (10 and 35%), on surface roughness, microstructure, porosity, mechanical properties, and corrosion behaviour in a simulated body fluid solution. The use of as-fabricated 3D structures will enhance the offering of more complex customised implants without the need for an additional surface finishing process.

From the results presented above, the following conclusions can be drawn:From XRD results, it can be concluded that the 3D RTCO 316L stainless steel structures retain the austenitic phase after the L-PBF process, similar to the fresh powder.The measured surface roughness (Ra) for both 10% and 35% density 3D RTCO structures is similar and falls within the range (~20 µm) of the fresh powder. This surface roughness is attributed to partially unmelted 316L SS powders present on the surface of the 3D structures.The density of the RTCO structures affects the distribution and morphological characteristics of the volumetric pores. The 3D RTCO structures with higher density (35%) exhibit smaller, more spherical, and fewer pores compared to the lower-density (10%) structures. Both types of structures contain pores resulting from a lack of fusion and keyhole defects.As-fabricated, non-surface finishing L-PBF-316L RTCO structures (10% and 35%) present a passive corrosion behaviour but with a lower pitting potential compared with the bulk polish L-PBF 316L SS. The non-surface finishing RTCO structures’ pitting potential is alike and compatible with a surface with identical roughness and passive film nature.The low porosity content has no obvious impact on the mechanical properties of the 3D RTCO structures, and the values obtained are of the same order of magnitude as the modulus of elasticity and yield stress of the bone tissue and are in line with the Gibson and Ashby model of an open cell structure.

In light of the results presented, further research is suggested to understand the effects of roughness on the biological response. Based on the scarce literature on the biological interaction of additively manufactured 316L SS implants, it is expected that the in vitro corrosion stability observed and the surface roughness obtained will provide information on the biocompatibility and cytocompatibility characteristics of these 3D structures without surface finishing, which are fundamental for the development of AM in biomedical implant applications.

## Figures and Tables

**Figure 1 jfb-15-00313-f001:**
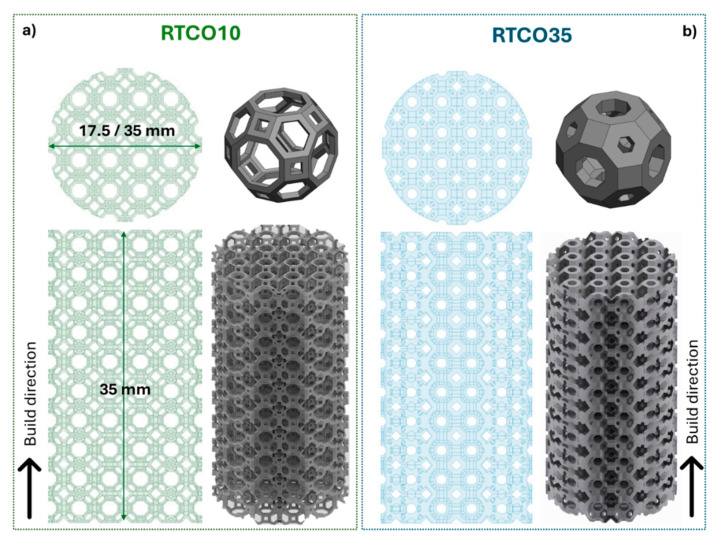
Design of the 3D structures with a RTCO lattice with top and front view perspective; (**a**) RTCO10, and (**b**) RTCO35.

**Figure 2 jfb-15-00313-f002:**
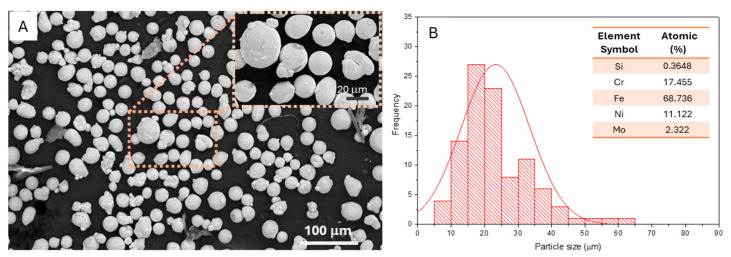
Scanning electron microscope (SEM) image of fresh 316L stainless steel powder (**A**); powder size distribution, and (**B**) chemical compositions obtained by EDS analysis.

**Figure 3 jfb-15-00313-f003:**
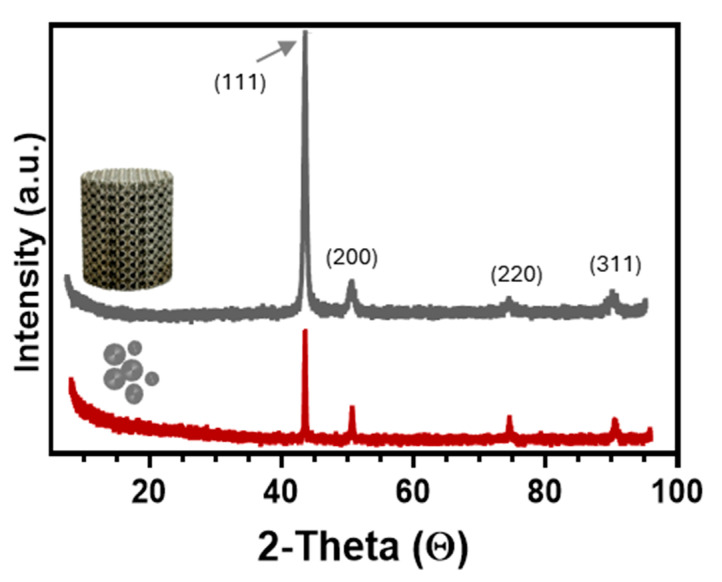
The XRD patterns of the fresh 316L SS powder and L-PBF 3D RTCO10 316L SS structure.

**Figure 4 jfb-15-00313-f004:**
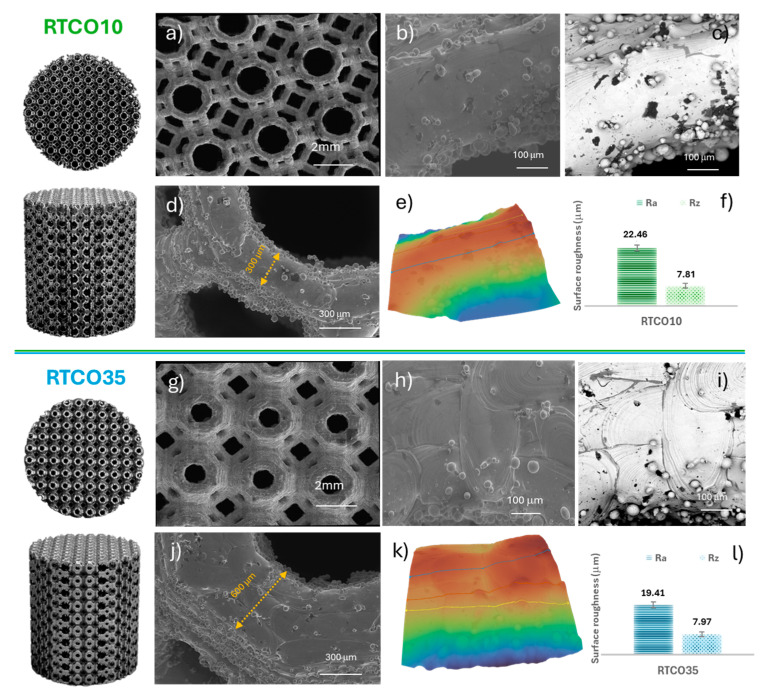
Scanning electron microscope (SEM) secondary electron images ((low-magnification, (**a**,**g**), high magnification, (**b**,**h**,**d**,**j**)) and backscattered images (**c**,**i**) for RTCO10 and RTCO35, respectively); and surface roughness and Ra and Rz of RTCO10 (**e**,**f**) and RTCO35 (**k**,**l**).

**Figure 5 jfb-15-00313-f005:**
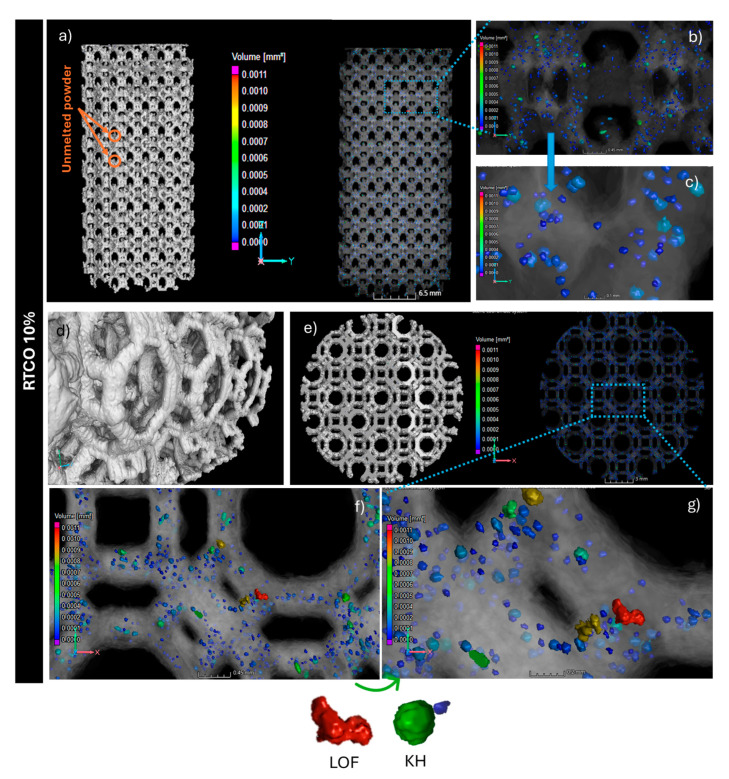
Porosity analysis of RTCO10 structures produced by L-PBF. CT slice image showing unmelted powder inside the cells (**a**,**d**). The porosity can be seen in an X-ray image and visualised and quantified in different ways from CT data, shown in a 3D side view (**b**) and magnified (**c**) or top view (**e**) and magnified (**f**,**g**). It detected a lack of fusion (red) and keyhole pores (green).

**Figure 6 jfb-15-00313-f006:**
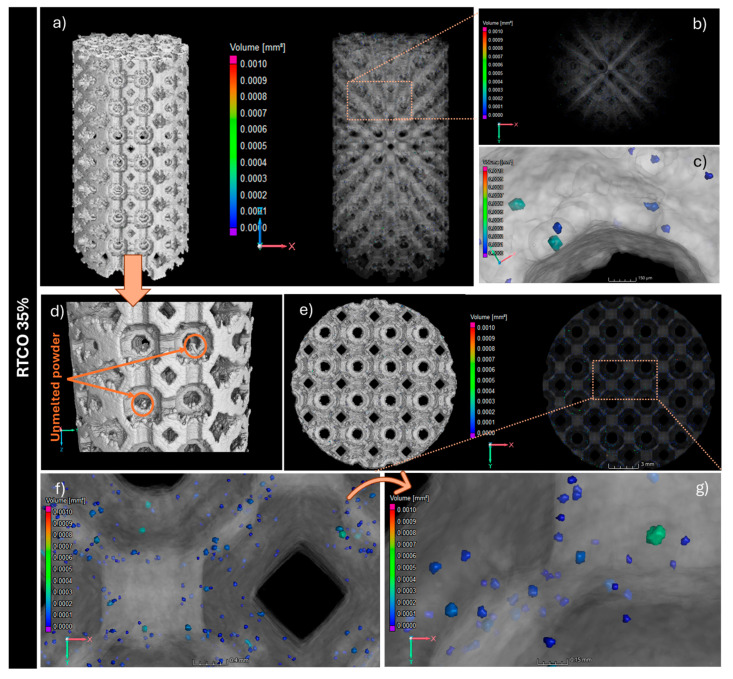
Porosity analysis of RTCO35 structures produced by L-PBF. CT slice image showing unmelted powder inside the pores (**a**,**d**); The porosity can be seen in an X-ray image and visualised and quantified in different ways from CT data, shown in a 3D side view (**b**) and magnified (**c**) or top view (**e**) and magnified (**f**,**g**).

**Figure 7 jfb-15-00313-f007:**
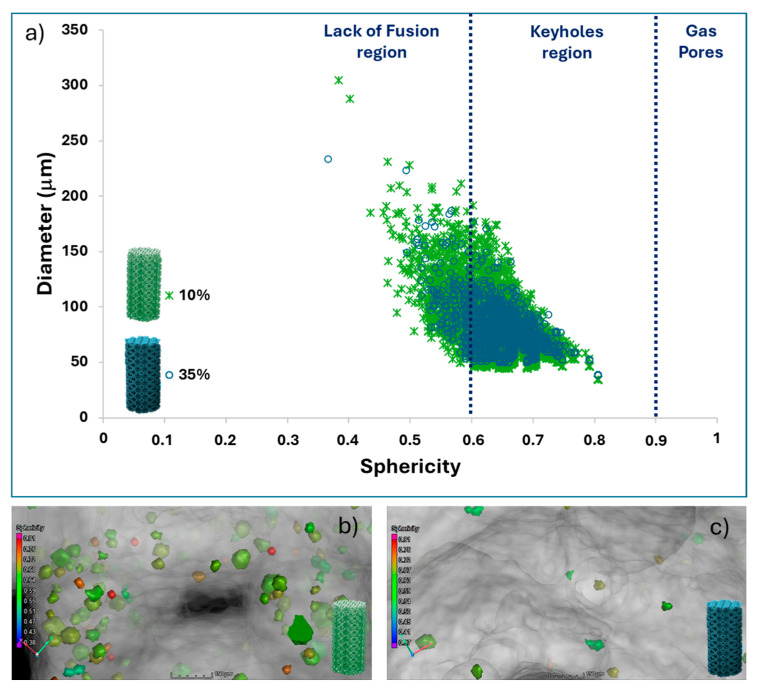
(**a**) Dependency of sphericity and pore diameter of RTCO10 and RTCO35. CT scan images as a function of sphericity; (**b**) RTCO10, and (**c**) RTCO35.

**Figure 8 jfb-15-00313-f008:**
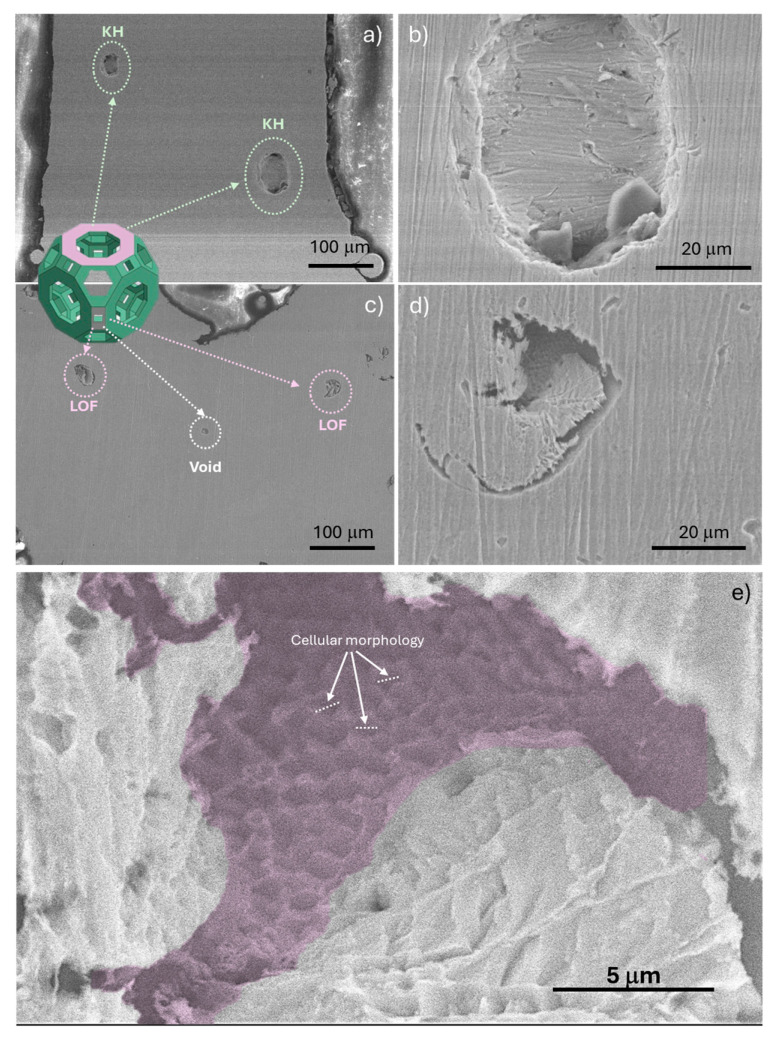
SEM images of 3D RTCO surface structures, viewed perpendicular (**a**,**b**) and parallel (**c**,**d**) to the building direction, highlighting a magnified region with cellular structure morphology (**e**).

**Figure 9 jfb-15-00313-f009:**
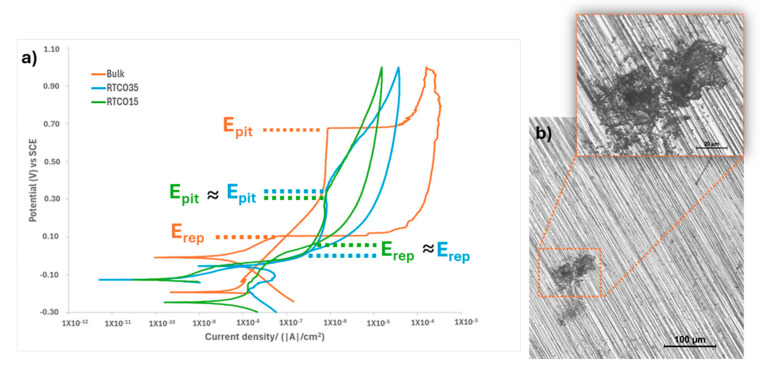
(**a**) Potentiodynamic polarisation curves for L-PBF structures with different densities; Bulk, non-surface-finishing RTCO10 and RTCO35 structures, immersed in a simulated body fluid (SBF) solution at 37 ± 1 °C; (**b**) Optical micrograph of pitting corrosion observed in polish bulk L-PBF.

**Figure 10 jfb-15-00313-f010:**
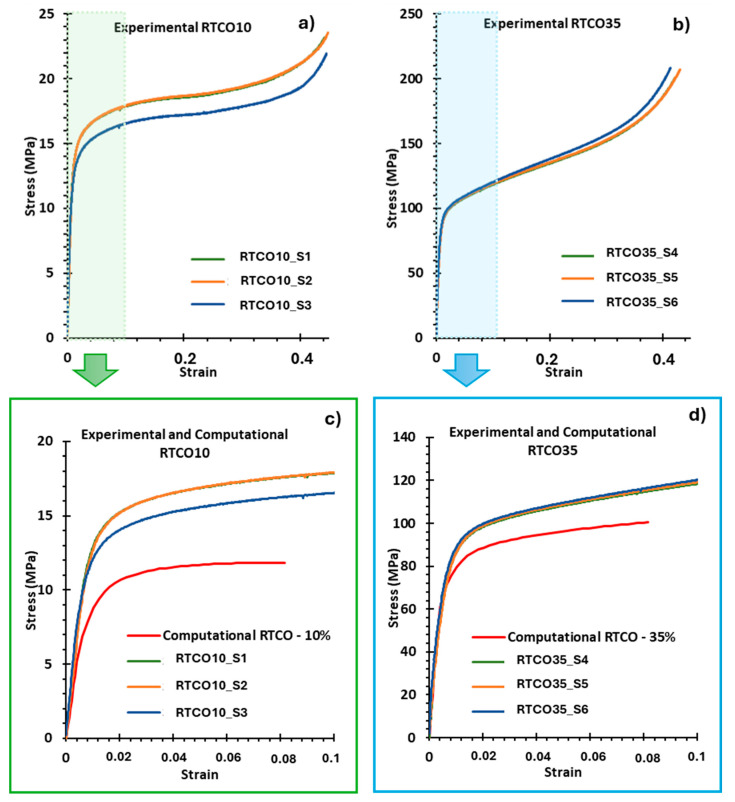
Experimental stress-strain curves (**a**,**b**) and comparison among the experimental and computational stress-strain curves (**c**,**d**) for RTCO10 and RTCO35 structures, respectively.

**Figure 11 jfb-15-00313-f011:**
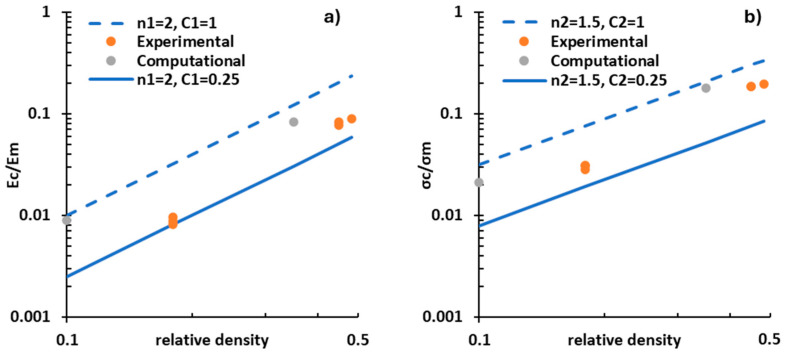
Log-log plots of (**a**) relative modulus and (**b**) relative yield stress as a function of the relative density.

**Table 1 jfb-15-00313-t001:** Relative density measured by the weight method of 316L SS 3D RTCO structures.

3D Structures	RTCO10_S1	RTCO10_S2	RTCO10_S3	RTCO35_S4	RTCO35_S5	RTCO35_S6
Theoretical Relative Density (%)	10	10	10	35	35	35
Real Relative density (%)	17.96	17.94	18.02	45.05	45.08	48.30

**Table 2 jfb-15-00313-t002:** Young’s modulus of experimental and computational RTCO structures.

RTCO	10_S1	10_S2	10_S3	10% Computational	35_S4	35_S5	35_S6	35% Computational
Young’s Modulus (MPa)	1729	1600	1899	1273	16,229	15,208	17,498	16,157
Average (MPa)	1743				16,310			
Yield stress (MPa)	13.9	13.8	12.7	9.4	84.0	83.8	87.7	79.7
Average (MPa)	13.5				85.2			

## Data Availability

The original contributions presented in the study are included in the article, further inquiries can be directed to the corresponding authors.

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
