# Peer review of "Mechanical and Corrosion Behaviour in Simulated Body Fluid of As-Fabricated 3D Porous L-PBF 316L Stainless Steel Structures for Biomedical Implants"

_jfb, 2024, doi:10.3390/jfb15100313_

Round 1
Reviewer 1 Report
Comments and Suggestions for Authors
The article is interesting, but more work is needed before it can be published. I have the following remarks:
1. line 33-34: The second part of a sentence is not corect in my opinion. "passive corrosion behaviour SBF solution"?.
2. line 87-91: What specific surface engineering methods do you mean? What chemical reagents and materials?
3. line 97-99: Without conducting biological tests you won't know this.
4. line 109-110: Please revise.
5. line 128: "with a layer thickness of 50 m, a spot size (hatch distancing) of 130 m". Please check units.
6. line 166: What was the step and step time?
7. line 190: Was the SBF solution purchased or prepared in a lab? If purchased, please indicate company, and if prepared please state how (literature reference)?
8. Why were the OCP results not presented?
9. Line 226: please check corectness of the phrase "attached for a few particles".
10. Fig. 2. What about carbon content? What is the source of the Al appearance?
11. Fig. 3: In the case of RTCO10 a shift of the (111) peak towards higher angles can be observed, compared to the fresh powder. Does this affect the properties? Please explain. Is the (111) pick also shifted for RTCO35?
12. line 299-300: This should be a separate sentence.
13. line 308: It's rather surface topography than surface roughness.
14. line 311: should be "GFP".
15. line 322: Should be (Figure 5fg).
16. Why are the results from Fig. 6 not described in the text?
17. Fig 5 and 6: "CT slice image showing unmelted powder inside the cells (a and d)". Can you mark it on the photos?
18. line 384: "a fine cellular morphology with an average cell size of approximately 0.76 ± 0.30 μm was observed". Please indicate on which Fig.
19. line 411: What does CP stand for?
20. Fig. 9: What about Ecor, icor, Rpol values? They are also important.
21. line 416: I don't understand the phrase "passive window"? Do you mean the passive state potential range?
22. Epit of The RTCO10 and The RTCO35 are almost two times lower compared to bulk material. Please explain why.
23. What Erep value was obtained for the bulk material?
24. lines 422-424. This sentense is not clear in my opinion. Please revise.
25. lines 437-439: Please indicate appropriate figures for this description.
26. line 447: Should be RTCO10_S3.
27. It is worth mentioning why RTCO35 has greater tensile strength than RTCO10?
28. lines 481-483. I think the real relative density doesn't matter in this case. For example, S4 and S5 have similar real relative density and a difference of about 1000 MPa in Young's modulus. I believe there must be other factor.
29. lines 500-510: What follows from this data?
30. line 526: Do you mean "of the fresh powder" diameter?
31. lines 543-548: A complex sentence, not clear, please revise.
Comments on the Quality of English LanguageThe text should be thoroughly checked by a native English speaker. A few sentences need to be rewritten.
Reviewer 2 Report
Comments and Suggestions for Authors
The manuscript Mechanical Behavior and Biodegradation in Simulated Body Fluid of As-Fabricated 3D Porous L-PBF 316L Stainless Steel Structures for Biomedical Implants is worth publishing after minor improvements
The article does not describe Biodegradation; I suggest changing the title to Mechanical and Corrosion Behavior .....
The introduction should be shorter; there are no cytotoxicity studies of manufactured materials so there is no need to mention it in the introduction (line82-92)
for the implants are very important: biocompatibility and mechanical or structural compatibility please explain what kind of implants you have in mind for AM parts
please add the data of 316L SS powder: producer, granulation, chemical analysis, and the thickness of oxide film on the surface of the initial powder (XPS, XRD, or SIMS)
Please correct the units in line 127
You may add the reference: Salama, M.; Vaz, M.F.; Colaço, R.; Santos, C.; Carmezim, M. Biodegradable Iron and Porous Iron: Mechanical Properties, Degradation Behaviour, Manufacturing Routes and Biomedical Applications. J. Funct. Biomater. 2022, 13, 72. https://doi.org/10.3390/jfb13020072
Round 2
Reviewer 1 Report
Comments and Suggestions for Authors
The article has been improved, but I still have some comments on the authors' responses:
Regarding comment 2: Thank you for the description but please specify it the article, which chemical reagents or materials can have negative contribution to the environment. I cannot find these reagents or materials in reference [13].
Regarding comment 6: Great, but why haven't you included these parameters in the text?
Regarding comment 8: Ok but OCP was performed before potentiodynamic tests. If so, please describe it in the methodology.
Regarding comment 17: The description on the pictures is barely visible.
Regarding comment 18: Thank you but I meant: indicate Figure 8e in the sentance in your article. In this case the description on the Fig. 8e is also barely visible.
Regarding comment 19: Ok, all clear, but the full name of the abbreviation should be provided in the text.
Regarding comment 20: Why didn't you use the greek letter instead of microA/cm2? We do not give the current value with a minus sign. Moreover, the values ​​on the Logi axis on the Fig. 9 are not correct.
Regarding comment 22: I meant, give the explanation in the article.
Regarding comment 23: Please include this Erep value and information in the text and explain the differences in Erep between bulk and RTCO materials. Is the reason the same as for Epit?
Author Response
Dear Review, please see the attachment.
